# Non-Reciprocal Frequency Contributions from the Active Medium in a Ring Laser

Alexander A. Velikoseltsev [1,*], Karl Ulrich Schreiber [2,3,4], Jan Kodet [2] and Jon-Paul R. Wells [3,4]

1 Department of Laser Measurement and Navigation Systems, Saint Petersburg Electrotechnical University "LETI", ul. Prof. Popova 5, 197376 St. Petersburg, Russia

2 Research Unit Satellite Geodesy, Technical University of Munich, Arcisstr. 21, 80333 Munich, Germany; ulrich.schreiber@tum.de (K.U.S.); jan.kodet@tum.de (J.K.)

3 School of Physical and Chemical Sciences, University of Canterbury, P.O. Box 4800, Christchurch 8140, New Zealand; jon-paul.wells@canterbury.ac.nz

4 Dodd Walls Centre for Photonic and Quantum Technologies, Dunedin 9016, New Zealand

* Correspondence: aavelikoseltcev@etu.ru

**Abstract:** Under ideal conditions, the optical path for the two counter-propagating beams in a square ring laser cavity is expected to be entirely reciprocal. This property, together with the absence of any moving parts in the gyro, makes ring lasers a very useful rotation-sensing device. For a typical aircraft application, a sensor stability of the order of 0.01 °/h and a resolution of 1 ppm is required. The demands for inertial rotation sensing in space geodesy are three orders of magnitude higher. Therefore, the perturbations from the presence of the active laser gain medium inside the cavity cannot be ignored. While these perturbations can be sufficiently contained in aviation gyros due to the much lower requirements, they cause a notable bias in large ring laser gyroscopes for the observation of the instantaneous rotation rate of the Earth. In this paper, we report on an improved model for bias stability from the presence of the laser gain medium in the gyro cavity of the large ring laser "G" at the Geodetic Observatory Wettzell. Typical values between 5 and 10 ppB are obtained over several months.

**Keywords:** inertial rotation sensing; ring laser; Earth rotation; laser gain medium

## 1. Introduction

Ring laser gyroscopes strapped down to an aircraft provide attitude control and, at the same time, autonomous navigation capabilities for as long as each flight takes [1]. These gyros are reset right before take-off and the accumulated positioning error after a 14 h flight is less than 2 km. In terms of specifications, this is well within the permissible inaccuracy requirement at the parts-per-million level. For a ring laser gyroscope, rigidly strapped down to Earth, for use in geophysical monitoring, we require a resolution that is about a factor of one thousand higher and that provides extreme sensor stability over many weeks [2]. Due to the large mass of the Earth, the geophysical signals of interest are very small, barely exceeding a rotation rate of $1 \times 10^{-12}$ rad/s. Increasing the area enclosed by the beam path of an optical Sagnac interferometer to 10 or more m$^2$ provides the necessary sensitivity. Upscaling alone, however, is not the full answer to the problem, because a larger instrument weakens the mechanical rigidity at the same time and thus compromises the crucial sensor stability significantly [3]. The "G" ring laser at the Geodetic Observatory Wettzell (Germany) has a rigid monolithic structure, encloses an area of 16 m$^2$ and currently provides the best compromise between sensor stability on one side and sensitivity on the other. A typical geophysical measurement series of this single-component gyroscope is shown in Figure 1 with the mean value subtracted. It clearly resolves polar motion and tidal signals below 10 parts in 10$^8$, emphasizing the exceptional resolution and stability of this gyroscope.

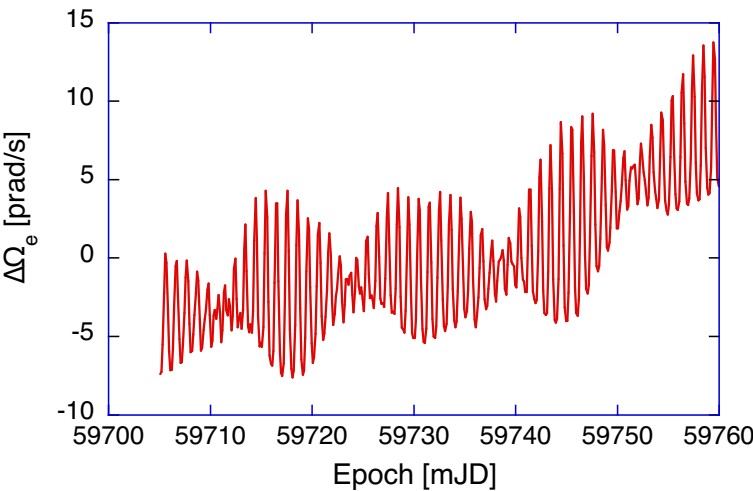

**Figure 1.** Example of a typical ring laser recording of Earth's rotation from the strapped down G ring laser. The measurements with the mean value of Earth's rotation subtracted are corrected for backscatter coupling and the null shift offset. They show a number of geophysical phenomena with good fidelity, but small sensor-related discrepancies remain.

Although not perceptible to us, the rotation rate of the Earth is not constant. There is a complex pattern of polar motion (precession, nutation) from the gravitational attraction of the sun and moon, the interaction of the fluids of the Earth (atmosphere, hydrosphere, cryosphere) and the solid part of the Earth, where the figure axis does not coincide with the axis of the moment of inertia [4,5]. Over the last two decades, we have developed a detailed model for the observed signals of a large ring laser [3], based on the properties of the sensor itself, the orientation of the gyro on the Earth surface and the perturbations of the rotation rate of the Earth. The example shown in Figure 1 was taken during May and June in 2022 with the mean value of $\Omega_e$ subtracted for clarity. Systematic sensor biases from backscatter coupling of the two laser beams in the interferometer [6], the null shift bias [7] and local sensor tilts and scale-factor corrections [8] have already been applied. One can clearly see a pattern with nodes and anti-nodes and a period of about 14 days as the moon progresses along its orbit. The diurnal wobble results mostly from the gravitational attraction of the moon and the small distortions from a clean sinusoidal pattern are caused by the experienced local tilt effects caused by solid-Earth tides. Finally, there is a pronounced upward trend towards the right side, which is induced by the Chandler motion [9]. In order to obtain all these signals correctly, it is important to ensure that the interferometer is unbiased and stable in the long term. This means that ideally there has to be no non-reciprocal frequency contribution larger than a small fraction of 1 µHz on the two counter-propagating laser beams in the ring laser cavity. This requirement corresponds to a relative frequency instability of $\Delta\Omega/\Omega \leq 2 \times 10^{-22}$. Therefore, it is important to take a closer look at the properties of the plasma in the gyroscope.

## 2. Systematic Errors in a Sagnac Interferometer

In our ring laser cavity, we use radio frequency laser excitation from a helium and neon gas mixture. In the cavity, there are two $^{20}$Ne and $^{22}$Ne isotopes present, each at 0.1 hPa partial pressure. On top of that, we add 9.8 hPa of helium in order to obtain a larger homogeneous linewidth from atom collisions in the gas mixture. This reduces the tendency to excite additional longitudinal laser modes and thus enables higher beam powers. Nevertheless, we are still operating relatively close to the laser threshold, since the free spectral range (FSR) is only 18.5 MHz and there are no mode-selecting devices permitted in the cavity in order to keep the losses low and hence the resolution of the gyroscope high. A Sagnac interferometer exploits the frequency difference of two intra-cavity excited single-mode laser beams, which travel around a ring cavity in opposite

directions on exactly the same optical path. Since both laser beams are generated from the same gain medium, it is inevitable that there are some perturbations introduced. The most prominent interaction is caused by the backscattering of one beam into the other and vice versa. We typically observe optical frequency pulling or pushing at the level of 11 ppm, corresponding to approximately 4 mHz out of a Sagnac beat note of 348.516 Hz. Our method for the correction of this effect is reported in [6].

Under perfect conditions, one would expect to find that both laser beams exhibit the same intensity because both beams traverse an identical optical path. Furthermore, there should be the same gain and the same loss for both laser beams. However, this is not the case. None of the gyros that our collaboration has built has ever shown equal beam powers. For *G*, the two beams in the cavity consistently differ in intensity by less than 1%. Other smaller and less sophisticated rings have even shown differences exceeding 5%. This asymmetry gives rise to a null shift bias and the corresponding correction is computed by applying a slightly modified version of the formalism introduced by [10] and also [7,11]. Typical values of this null shift correction are of the order of 2 ppm for *G* and are observed to be fairly stable over time. Figure 2 illustrates this typical operation scenario for a situation where the ring laser is operated under an ambient-pressure-stabilizing vessel to avoid compression effects on the gyroscope scale factor caused by changing atmospheric pressure patterns. The slope efficiency the 633 nm neon transition is too high to operate the gyroscope in a regime of stable intensity with a constant rf power setting of the exciter, hence we use a feedback circuit to servo the optical output power of the clockwise propagating laser beam. The other beam is thus not constrained. The top two panels in Figure 2 show the variability in the observed laser gain for each beam over the first week of operation after they are switched on. While the intensity of the controlled beam is regulated to remain constant, the uncontrolled beam reveals some variation, indicating a changing differential loss between the two laser modes during the first two days as the laser operations settle down. These changes are at the level of 1 part in a billion and indicative of small asymmetry in the Sagnac interferometer. We note, however, that this effect is not caused by backscatter coupling, which has been derived and corrected already.

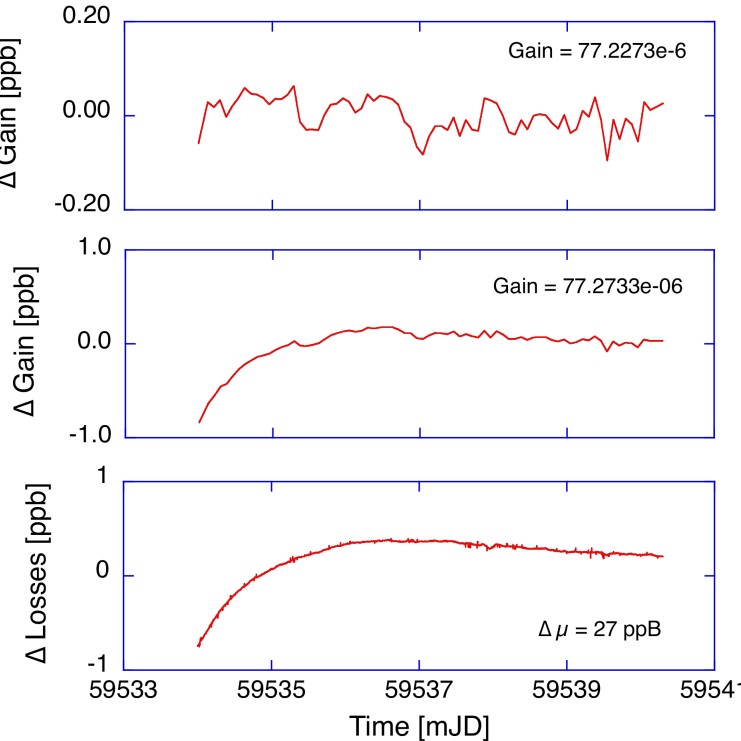

**Figure 2.** The variation in gain for the controlled beam (**top**), for the uncontrolled beam (**middle**) and the difference in losses (**bottom**).

While we observe such operational effects, the cause for this change in gain or loss for two laser beams traveling on exactly the same optical path has not yet been satisfactorily identified. We suspect some very small non-reciprocal interactions inside the coatings of the turning mirrors in the cavity though. For our analysis, we usually estimate the gain and the loss of each beam as a function of time and estimate and apply both the backscatter correction and the null shift bias with frequently updated gain and loss values. This improves the sensor correction, but still leaves us with some signal variability that cannot be explained in this way.

The temperature in the ring laser laboratory is very stable with a day-to-day variation of no more than 0.003 degrees. It dropped by 1° in the first year and then stabilized to ±0.1° in the next year with a signature following the seasons. There was no human intervention in that time. This undoubtedly causes some small variation in the cavity size, although the Zerodur base of the monolithic gyro achieved a residual thermal expansion coefficient of $\alpha \approx 5 \times 10^{-9}$. The result is a moderate drift in the optical frequency as the effective length of the cavity changes over time, and that in turn generates a variation in dispersion. In order to study the corresponding effect, we have introduced a monitoring scheme for the optical cavity frequency [8] based on an optical frequency comb, slaved to a hydrogen maser. This allows us to observe the optical frequency drift in the cavity continuously and with high resolution.

### 3. The Optical Frequency Observed in a Ring Laser Gyro

*G* has a square contour with four mirrors forming the cavity. This is equivalent to four output ports, from which we use one for the interference of the two laser beams. Two other ports are taken for the determination of the intensity of each of the two laser beams. There is one port left for the determination of the optical frequency excited in the cavity. Figure 3 illustrates the setup schematically. Typical beam powers are in the regime around 25 nW. Directly beating this signal with an optical frequency comb set up for a wavelength of 633 nm does not provide enough beam power for the mixed beam to be directly compared to the H maser. Therefore we have to include a transfer laser in the setup.

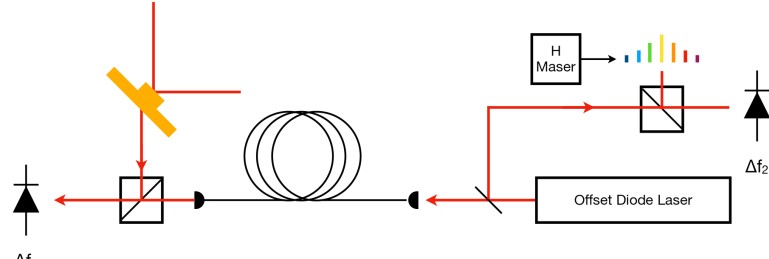

**Figure 3.** Block diagram of the optical frequency detection. The output of a CW diode laser is split into two beams. One beam obtains the the beat frequency with the ring laser, the other beam is beating with an optical frequency comb, which in turn is referenced against a H maser. From the two beat notes and the comb parameters, the optical frequency is derived with a resolution of ±1 kHz.

The output of a CW diode transfer laser is split into two beams. One of the beams is injected into a mono-mode fiber, which is routed into the temperature stable ring laser laboratory and fed to a free-space optical beat unit close to the ring laser. The weak output beam of the ring laser is superimposed and the beat frequency $\Delta f_1$ is detected by a sensitive photodiode and is used to lock the transfer laser to the ring cavity. The other beam of the transfer laser is fed to another beat unit on the optical table of the frequency comb. Another photodiode detects the beat frequency between this beam and one tooth of the comb and thus provides the beat note $\Delta f_2$. Together with the offset frequency of the optical frequency comb and its repetition rate, the optical frequency of the counter-clockwise beam in the ring laser is established continuously and produces an averaged value at every minute. Figure 4 presents the time series of the laser frequency in the G ring cavity.

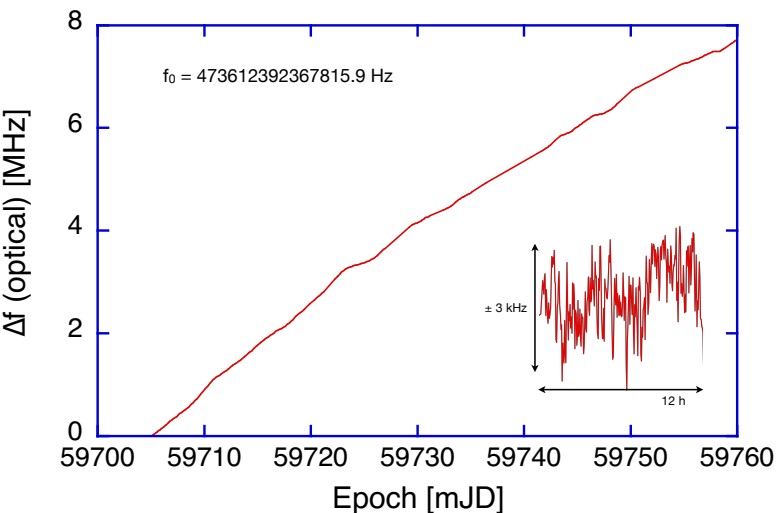

**Figure 4.** Display of the behavior of the optical frequency in the ring laser cavity averaged over 3 h for 55 days. The inset shows a typical section of 12 h duration averaged over 1 min.

One can see that the optical frequency is very stable. It drifts upwards slowly by 8 MHz over 55 days. The instability of the optical cavity has a standard deviation of 1.2 kHz over an averaging time of 1 min, which is illustrated in the inset of Figure 4. The observed drift is the expected response to the annual cycle of the ambient temperature in the ring laser laboratory, which dropped very smoothly by 0.16° over the entire duration of the measurement. The large mass of the Zerodur block smoothes this trend even further. A shrinking perimeter increases the optical frequency, which is consistent with the observation.

## 4. The Effect of The Optical Frequency Drift and Scale Factor Correction

The main objective of a Sagnac interferometer is the establishment of complete reciprocity of the two counter-propagating laser beams. This requires the absence of backscatter coupling, null-shift effects and high linearity in the laser beat note in response to the experienced rate of rotation. This latter condition is met, since Earth rotation only changes over a very small range. Backscatter and null shift can be sufficiently corrected by observing the intensities of the involved laser beams. It is usually assumed that laser gyros operate at the line center of the 633 nm transition. This is certainly true for small gyros, which have a large longitudinal mode spacing. For our large gyroscope, this condition no longer holds. The FSR is only 18.5 MHz and the homogeneous line broadening is large, due to the high gas pressure in the cavity. It suppresses neighboring laser modes over a range of $\pm 100$ MHz. That means that a small asymmetry from changing dispersion can be expected when the optical frequency changes in the cavity. The comprehensive theoretical formalism, developed by F. Aronowitz [1], which was later adapted to the application of large ring lasers, as summarized in [3], yields the following expression for the Sagnac beat frequency $\Delta f$ in the presence of differential pulling:

$$\Delta f = \frac{4A}{\lambda L}\left(1 + \frac{c}{2L}\frac{G}{Ku}\frac{Z_r'(\xi)}{Z_i(0)}\right),\tag{1}$$

where $A$ is the area enclosed by the two counter-propagating beams, $\lambda$ is the laser wavelength, $L$ is the ring laser cavity perimeter and $G$ is the laser gain. $Ku$ represents the Doppler broadening parameter, $Z_r$ and $Z_i$ are the real and imaginary parts of the plasma dispersion function and $\xi$ is the oscillation frequency detuning. The latter is defined as $\xi = (\nu - \nu_0)/Ku$ and accounts for the deviation in the oscillation frequency $\nu$ from the center frequency $\nu_0$ of the laser line, divided by the Doppler broadening parameter $Ku$. The

first part before the brackets in Equation (1) corresponds to the geometrical scale factor and is predominantly defined by the physical dimensions of the resonator. Any fluctuations in this value are usually defined via the area-to-perimeter ratio. Hence, the geometrical scale factor part can be described as

$$K_g = \frac{4A}{\lambda L}\left(1 + \delta\left(\frac{A}{L}\right)\right),\tag{2}$$

The part inside the brackets of Equation (1) characterizes the contribution of the active medium to the scale factor, due to the difference of the pulling induced by both oscillations. Since our large ring lasers operate on an equal $^{20}$Ne and $^{22}$Ne isotope mixture, and following [1,12], we can rewrite the plasma dispersion function parts from Equation (1) as

$$\begin{aligned}
Z_r(\xi) &= 2k_1\xi_1(\sqrt{\pi}\eta e^{-\tilde{\xi}_1^2} - 1) + 2k_2\xi_2(\sqrt{\pi}\eta e^{-\tilde{\xi}_2^2} - 1) \\
Z_i(\xi) &= k_1\left(\sqrt{\pi}e^{-\tilde{\xi}_1^2} - 2\eta + 4\eta\tilde{\xi}_1^2\right) + k_2\left(\sqrt{\pi}e^{-\tilde{\xi}_2^2} - 2\eta + 4\eta\tilde{\xi}_2^2\right),
\end{aligned}\tag{3}$$

where $k_1$ and $k_2$ are the fractional amount of each isotope, $\xi_1$ and $\xi_2$ are the detuning with respect to the $^{20}$Ne and $^{22}$Ne line centers and $\eta = \gamma_{ab}/Ku$ is the relative value of the homogeneous broadening to the Doppler broadening. The value of $Z_i(0)$ in Equation (1) hence can be obtained by substituting $\xi_1$ and $\xi_2$ with $\xi_{M1}$ and $\xi_{M2}$, which are the measures of the detuning from the maximum gain, located in the center between the two isotopes, whose line centers are offset by approximately 890 MHz. The homogeneous broadening can be approximated as $\gamma_{ab} = (10 + 59p_{He} + 29p_{Ne})$ MHz with $p_{He}$ and $p_{Ne}$ representing the partial pressure of He and Ne components of the gas mixture [13], while $Ku = \Delta\omega_D/2\sqrt{\ln 2}$, where $\Delta\omega_D$ is the FWHM Doppler width.

Another active medium contribution to the scale factor is differential pushing, which modifies Equation (1) in the following way [14]:

$$\Delta f = \frac{4A}{\lambda L}\left(1 + \frac{c}{2L}\frac{G}{Ku}\frac{Z_r'(\xi)}{Z_i(0)} - \frac{c}{2L}\frac{G}{Ku}\left(\frac{\xi}{\eta}\mathcal{L}(\xi)Z_i(\xi)\right)'\frac{\alpha(\xi)}{\beta(\xi) + \Theta(\xi)}\right),\tag{4}$$

where $\alpha$, $\beta$ and $\Theta$ are Lamb coefficients in the amplitude equations for the counter-propagating beams. They can be written as [10,11]

$$\begin{aligned}
\alpha(\xi) &= G\frac{Z_i(\xi)}{Z_i(0)} - \mu \\
\beta(\xi) &= G\frac{Z_i(\xi)}{Z_i(0)} \\
\Theta(\xi) &= G\frac{Z_i(\xi)\mathcal{L}(\xi)}{Z_i(0)},
\end{aligned}\tag{5}$$

Here, $\mathcal{L}(\xi) = [1 + \xi/\eta]^{-1}$ represents a Lorentzian function, $G$ is the gain and $\mu$ is the total cavity loss. Since all the parameters in Equation (4) depend on frequency detuning, the knowledge of the optical frequency drift, as shown in Figure 4, provides the necessary information for the active medium scale factor correction model. By taking the derivatives of the pulling and pushing expressions, which are signified by primes in Equation (4), substituting all known ring laser parameters (see Table 1), as well as the measured optical frequency drift, we obtain the corresponding pulling and pushing correction as a function of time, based on the drift in the optical frequency. Figure 5 illustrates the respective corrections. For optical frequencies close to the line center, the differential pushing term dominates, while the differential pulling term would dominate the region when the optical frequency is further away from the common line center of the two neon isotopes than about 100 MHz. In order to obtain the corrections as shown in Figure 5, we need to establish the initial offset of the optical frequency from the common line center. One would expect it to

be located within ±FSR relative to the center line, which implies single-mode operation. However, since the maximum for the two-isotope mixture is very flat and due to the wider homogeneous broadened lines for the over-pressured $G$, the frequency span of single-mode operation is found to be much wider than that in [15]. We obtain the best agreement when the offset is around +32 MHz, which is within reasonable limits.

**Table 1.** Parameters used in the differential pushing and pulling calculation.

| Parameter | Quantity | Estim. Uncertainty |
|---|---|---|
| gain | $7.8 \times 10^{-5}$ | 1 ppm |
| total losses | $5.3 \times 10^{-5}$ | 1 ppm |
| $T_{plasma}$ | 300 °K | 2% |
| offset from gain center | 32 MHz | 0.2% |
| $\gamma_{ab}$ | 448 MHz | 1% |
| $Ku$ | 771 MHz | 3% |

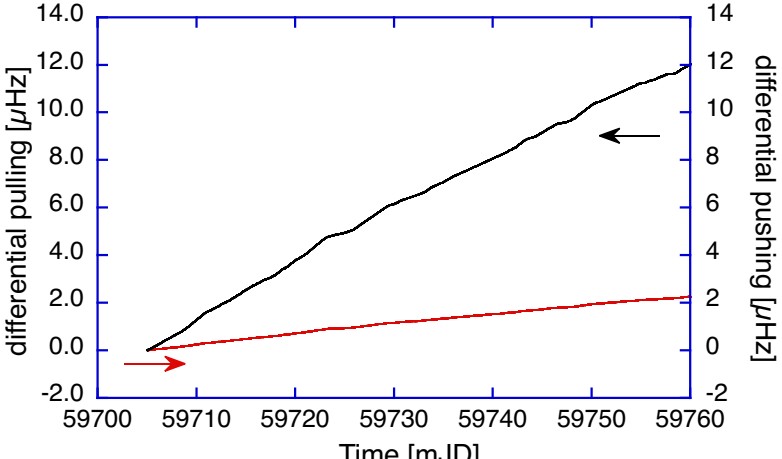

**Figure 5.** Display of the calculated corrections for the differential pulling and pushing terms. The obtained corrections are below 0.1 ppm and therefore relevant for high-resolution Earth rotation sensing. Notice the difference in scale between the two contributors.

Over the entire length of the observation, we obtain corrections at a level of less than 0.1 ppm, again about one order of magnitude smaller than the null shift correction. For the application of high-resolution inertial monitoring of instantaneous Earth rotation, this contribution matters and cannot be neglected. Figure 6 presents the residuals of the ring laser observation of $G$ over the same two months as shown in Figure 1, with the geophysical models for diurnal polar motion, Chandler and Annual wobble, solid-Earth tides and the variation in length of day applied after the ring laser beat frequency had already been corrected for instrumental effects, namely backscatter coupling and null shift.

The measurement residuals as well as the differential pushing and pulling corrections for a drifting optical frequency were converted to the corresponding rotation rate in units of prad/s. We resolve rotations to well below 1 prad/s after three hours of averaging. Apart from some remaining systematic effects, which are most likely caused by deficits in the recovered local tilt, our model extension accounts well for the remaining small upward drift in the ring laser observations. When we apply the corrections related to the drift in the optical frequency to the mostly unperturbed section between days 59,725 and 59,755, we obtain standard deviations for the measured rotation rate of 0.45 prad or 8 parts in $10^9$ over a whole month.

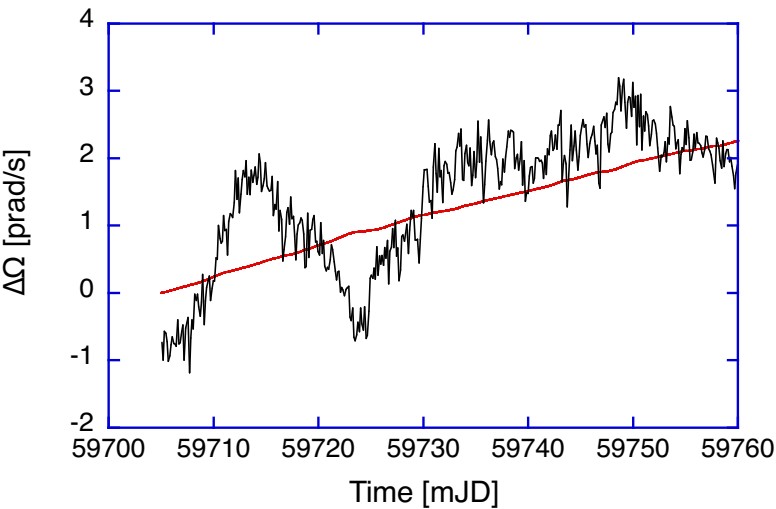

**Figure 6.** The observed ring laser residuals experience a small upward drift, which can be compensated by the inclusion of pushing and pulling terms related to the drift in the optical frequency in the gyroscope correction model.

### 5. Discussion and Conclusions

A large ring laser gyroscope, strapped down to Earth at mid-latitudes, provides Earth rotation measurements with extremely high resolution. These measurements become scientifically relevant when it is possible to reach the domain below 1 part in $10^8$, because the sensor then can resolve the tiny variations in the length of the day. This will be a relevant contribution to the geosciences because the observations provide constraints for models of global mass transport phenomena. These constraints help to distinguish valid models from those which are not applicable. The Earth system is marked by very complex interactions, composed of many contributors, and cannot be modeled analytically. The application of large ring laser gyros, strapped down to the body of the Earth, have an advantage here, because they operate continuously and autonomously. However, the requirements for an inertial sensor are very tough. Measurement resolution, although very necessary, does not provide the full answer. Above all, it is the sensor stability that matters. Even a small amount of sensor drift at the order of 1 part per billion per day is enough to make seasonal effects of mass transport unobservable. The small dispersive effect of minor laser oscillation frequency detuning that we have examined in our work provides a considerable improvement to the stability of our sensor. The mechanical structure of the *G* ring laser in Southern Germany is, in principle stable, enough to reach this goal. However, despite all care taken, it is not an ideal instrument. Apart from the well-known systematic bias mechanisms, such as backscatter coupling, it also persistently shows differences in the intensity of the two counter-propagating laser beams. The continuous observation of the optical frequency of the gyroscope also reveals a slow but steady drift as the temperature in the laboratory slowly goes through its annual cycle. The variation in the ambient atmospheric pressure has been actively controlled by placing the gyroscope under a pressure-stabilizing vessel, so that the compression from the atmospheric loading of the body of the ring laser remains constant at all times. We used the Lamb formalism in the model of F. Aronowitz to account for the difference in loss between the two laser beams in the cavity and the changing scale factor evidenced by the observed changing optical frequency. Both effects lead to a small systematic error correction, which can explain the observed residual drift well. Apart from that, we still observe some systematic signatures in the ring laser residuals. This is shown in Figure 6, where all known influences are removed. The apparent upward trend in this graph could be explained successfully by the dispersion correction, derived from the drift in the optical frequency in the laser cavity. The oscillation in the first half of the measurement series is still unexplained. Our gyroscope is a single-component sensor, which is highly sensitive to the north–south tilt. A tilt of 3 nrad

produces a shift in the interferometer beat note by one part per billion. This is enough to make the net effect of global mass transport unobservable. In order to capture the variation in tilt, we have several tiltmeters with a resolution of 1 nrad placed on the Zerodur disc of the gyroscope, monitoring any changes in tilt in the north–south direction. Although they mostly agree fairly well, the observed discrepancies among them are large enough to explain the magnitude of the remaining deviations in Figure 6, but we cannot reach perfect agreement. Another problem would be local rotation experienced by the sensor. Taking the first slope in Figure 6 and integrating the rate of rotation would result in an angle of approximately 34 nrad per day. This appears to be too large to be the cause of the observed deviation. There are still some geophysical signals, which are not yet included in our reduction model, such as the effect from ocean loading and zonal tides. Since they are periodic, they would show up in the spectrum. At this point in time, we cannot resolve such a signal contribution. The steep drop between day 59,720 and 59,730 is caused by local tilt effects, which have not been adequately captured by our tiltmeters. These devices are not only sensitive to tilt: the pendulum is also influenced by gravitational attraction and it is not simple to untangle the deformation part, namely tilt, from the attraction component (gravity). So, while the tiltmeters present a combination of both effects, a ring laser gyroscope is entirely insensitive to mass attraction and only reacts to a change in the projection of the rotation vector onto the ring laser plane. In summary, we have identified and removed the dispersive effects of the optical frequency drift in the gain medium of our large helium–neon gas laser gyroscope. The excellent stability of the monolithic cavity of the ring laser *G* allowed us to measure the optical frequency of the excited laser mode with a resolution of 1.2 kHz continuously over two months. The precise measurement of the ring laser's operational parameters and the application of the Lamb formalism allowed for the identification and retrieval of a persisting drift signal in the gyroscope operation. This takes us one step closer to a stable sensor operation good enough to extract seasonal variations in global mass transport phenomena.

**Author Contributions:** All authors contributed in a meaningful way to the development of the sensor setup over an extended period of time. K.U.S. and J.K. carried out the measurements of the rotation rate and the optical frequency, A.A.V. and K.U.S. provided the analysis and wrote the manuscript, J.-P.R.W. co-wrote the manuscript. All authors have read and agreed to the published version of the manuscript.

**Funding:** This research received no external funding

**Institutional Review Board Statement:** Not applicable.

**Informed Consent Statement:** Not applicable.

**Data Availability Statement:** The data will be made available by the authors upon reasonable request.

**Acknowledgments:** This work has been carried out in the framework of the research program of the *Research Group Satellite Geodesy*. We acknowledge funding from the Federal Agency of Cartography and Geodesy and the Technical University of Munich.

**Conflicts of Interest:** There are no competing interest in this work.

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
