# Peer review of "Non-Reciprocal Frequency Contributions from the Active Medium in a Ring Laser"

_photonics, doi:10.3390/photonics10111241_

Round 1

Reviewer 1 Report

Comments and Suggestions for Authors

The submitted manuscript reports bias stability of a gain medium for large ring laser gyroscopes (RLEs). The author(s) have long contributed to the area of RLEs. Authors report various data from their RLEs whose variations are relevant to various error sources such as Earth, Moon, and the gain medium of the laser. For example, Fig. 1 is for the error from geophysical phenomena, whereas Figs. 2 and 5 are for the gain medium. Figures 3 and 4 related to the annual temperature changes do not seem to be solely relevant to the gain medium. The manuscript is written more in a narrative tone for their usual research activities rather than hard work to overcome the problems. To general readers, the manuscript looks easygoing without a serious point, even though the authors discuss it. In that sense, it should’ve been more focused on the main topic of the gain medium. The related data, e.g., Fig. 5 should be compared with others if possible and discussed with the level of target sensitivity or stability to be achieved. For this, I think, certain physical limitations or application goals of RLEs may be shown to fit the purpose of this paper as mentioned in the Abstract. For these, detailed numbers are needed. The importance of 0.1 ppm in Fig. 5 should be explained or discussed with technological difficulties or a theoretical limit. All equations are from other papers. I wonder if these can be the solutions to the gain medium issue with backscattering lights. Then, the novelty of the paper is severely weakened. As long as the goal of the manuscript is in the stability of RLEs, detailed solutions (theory or method) on how to overcome them must be represented, even though Fig. 5 shows the related improvements. Based on the comments above, I cannot recommend publication of the submitted manuscript in Photonics. A major revision is required.

Author Response

Dear reviewer:

  1. We believe that all figures are relevant for the paper. Figure 1 is a necessary introduction to the current state of the art of large ring laser observations. It emphasizes the multitude of geophysical signals that are contained in the ring laser measurements. We wish to make the point that the observation of interest is the behavior of the system earth. The ring laser is the tool that needs to satisfy very stringent specifications, which are hard to achieve. In order to get access to the low frequency domain of the geophysical signals, we have to improve mostly the stability of our sensor. By removing the dispersive effects we achieve a considerable improvement, with all other larger bias errors removed already. Figure 2 provides insight into the bias causing asymmetry in the ring laser operation. Both beams share a common path in the cavity, yet the losses are different and this effect is mostly independent of the ambient temperature. This is certainly relevant to our presented work. We have modified this section in order to make this clearer. Figure 5 contains the result from our calculation, namely the calculated differential pulling and pushing contribution. This is the main result of our manuscript.
  2. Figure 3 and figure 4 explain how the experiment to obtain the optical frequency of the Sagnac interferometer was set up and what the observed variation of the optical frequency in the cavity is. To set this measurement technique up is a major undertaking and not trivial, since we are dealing with very low beam powers. Figure 4 depicts the trend of the optical frequency together with the resolution of our measurement system. We believe that these figures are highly relevant and of excellent quality. They provide the input for the model section. Neither figure 3 nor figure 4 deals with temperature changes, although we certainly agree that the optical frequency drifts because of thermal expansion of the Zerodur body.
  3. We have checked our document and added a couple of clarifying remarks throughout the document. In this paper we report on the removal of a continuous small drift at the level of 2 parts per billion per day, which requires an optical frequency comb, an elaborate locking scheme and high resolution error correction for all the other non-reciprocal frequency contributions, which is built on our previous work and presented in the references. Furthermore, we have clearly explained how we processed the data. We do not understand why the reviewer thinks that there is no serious point made.
  4. We have identified differential dispersive effects as the source of the bias drift that we observe with our gyroscope (see fig. 5 and 6). In order to do so, we were required to track the optical frequency over several months and we have devised the necessary instrumentation to do so. The result is a considerable reduction in sensor drift, which in turn allows us to examine geophysical signals at much lower frequencies, approaching 1 microHz or expressed in periods, 14 days. For an optical interferometer, this corresponds to a major achievement in stability. We have extended the discussion section to make the relevance of our work clearer. This concerns the stability and resolution. To our knowledge there are no other results in the literature of comparable resolution.
  5. Figure 5 shows the computed corrections, which are at the level of 1 ppb and not 0.1 ppm as the reviewer writes. This is significantly below the level of the backscatter and null shift contribution, which was removed before the analysis of the differential pulling/pushing was performed. Since that is described in the references, we did not make this a topic in this paper. The reviewer is right, there are limitations. These are given by the losses in the cavity and Brownian coating noise. Both experimental limitations are related to the mirror coatings. At this point in time, this is a hardware limit, which would require a new technical development to resolve. However, this is beyond the scope of our paper.
  6. Figure 5 shows the results from the model calculations and plots the combined result from figure 5 onto the residuals of our ring laser observations in figure 6, which is the red line. It convincingly shows that the drift goes away, which is the main objective of our development effort and the main scope of this manuscript. We believe that this is a very convincing result.
  7. We have reported on the measurement of the optical frequency in order to derive the necessary input parameters for the calculations of the differential corrections of dispersive effects, based on the existing models following the well established Lamb formalism. We reiterate, we do not deal with backscatter modeling here that is covered in the references 6 - 8 with sufficient resolution.
  8. Upon request of the reviewer, we have edited our manuscript considerably in order to make our message clearer.

Reviewer 2 Report

Comments and Suggestions for Authors

This paper proposed an improved model for the bias stability from the presence of the laser gain medium in the gyro cavity of a large ring laser,and applied this model in “G” at the Geodetic Observatory Wettzell. The typical values are smaller than 10 ppB. The authors showed sufficient results supporting their conclusions. Therefore, I recommend to accept this manuscript without any corrections.

Author Response

Dear reviewer, thank you very much for your judgment.

Reviewer 3 Report

Comments and Suggestions for Authors

In this paper, the authors discuss the sources of bias and uncertainty in their large ring laser gyros, and how they are overcome. They first recall previously described aspects: null shift due to backscattering and residual intensity difference between the counterpropagating beams, scale factor variation with temperature. Then they present a new method to measure the optical frequency of the ring laser based on a frequency comb, and how it can be introduced in a Lamb equation model to compensate for residual frequency pulling and pushing effects. They show that this model reasonably well explains the slow drift in their data after correcting for all other known sources of bias and geophysical effects. I think the paper is interesting, well-written and scientifically sound and I recommend its publication in its present form. I have a few remarks that might help improve the paper :

-page 2 line 62: I disagree with the statement Delta Omega/Omega<2e-22: in my opinion, since Delta Omega refers to the stability of the beat note it should be compared to the beat note frequency (350Hz, leading to Delta Omega/Omega on the order of 1e-9), not to the absolute optical frequency (which leads to 1e-22)

-page 4 line 130: I wonder about the effect of the monomode fiber on the stability/accuracy of the frequency measurement as it might add some phase/frequency noise from length and refractive index fluctuations…

-page 6 lines 188-192: I wonder why the absolute position of the center line cannot be determined independently, from known properties of He-Ne. Indeed the +32MHz offset which is chosen to best agree with the data sounds a bit surprising (although not unreasonable), and assumes that the observed drift on figure 6 is indeed due to frequency pulling/pushing and not to some undescribed geophysical effect… I would be interesting for the reader to know how the pushing/pulling effect changes with this offset.

Author Response

Dear reviewer:

  1. This is indeed a matter of taste. We agree with the reviewer statement, but we like to put that in a wider perspective. A Sagnac interferometer requires the two counter-propagating beams to be entirely reciprocal. Any small displacement of any of the two optical frequencies introduces a bias, which shows up as a drift term in the beat note and this is of course not a rotational signal. However, it originates from a slight pulling or pushing of any of the two optical beams. I personally prefer to view this pushing of the beam (delta omega) relative to the optical frequency of the beam (omega), because it is there where the offset originates and the bias process happens. Since it comes down to the beat note with exactly the same amount because it is a non-reciprocal effect, your viewpoint is also right, but there is no causal relationship in it. Therefore I prefer our view.
  2. The fiber link is actively delay compensated. We did not specifically mention that, because the effect on the frequency evaluation for the locking scheme is negligible. If we would be looking at the phase however, it would not be negligible.
  3. We do not exactly (to the MHz) know where the line center is. This depends on the details of the partial pressures of the neon isotopes. But we know that the line center is very flat, due to the large isotopic shift between 20Ne and 22Ne. Furthermore we operate where the ring chooses to lase. Due to the over-pressuring, we have a fairly large regime of +- 100 MHz where it could come on before it changes the longitudinal mode. Since the detuning is a sensitive parameter, we were essentially inferring the offset from the best agreement of our computed correction. We have added some statements to this effect to the manuscript. If the offset from line center is 0, the corrections would be too small to matter. If the offsets were somewhat larger, the correction would become significantly larger. So the best agreement value is fairly well defined.

Sincerely, Alex Velikoseltsev, Ulrich Schreiber

Round 2

Reviewer 1 Report

Comments and Suggestions for Authors

The submitted manuscript reports bias stability of a gain medium for large ring laser gyroscopes (RLEs). The author(s) have long contributed to the area of RLEs. Authors report various data from their RLEs whose variations are relevant to various error sources such as Earth, Moon, and the laser's gain medium. For example, Fig. 1 is for the error from geophysical phenomena, whereas Figs. 2 and 5 are for the gain medium. Figures 3 and 4 related to the annual temperature changes do not seem to be solely relevant to the gain medium. All raised issues are resolved in the revised version. Thus, I recommend the revised manuscript for publication in Photonics.